# Depletion of Paraoxonase 1 (Pon1) Dysregulates mTOR, Autophagy, and Accelerates Amyloid Beta Accumulation in Mice

**DOI:** 10.3390/cells12050746

**Published:** 2023-02-26

**Authors:** Łukasz Witucki, Hieronim Jakubowski

**Affiliations:** 1Department of Biochemistry and Biotechnology, Poznań University of Life Sciences, 60-637 Poznań, Poland; 2Department of Microbiology, Biochemistry and Molecular Genetics, International Center for Public Health, New Jersey Medical School, Rutgers University, Newark, NJ 07103, USA

**Keywords:** APP, amyloid beta, *Pon1*^−/−^5xFAD mouse model, N2a-APP_swe_ cells, Pon1, homocysteine thiolactone, Phf8, H4K20me1, mTOR, autophagy

## Abstract

Paraoxonase 1 (PON1), a homocysteine (Hcy)-thiolactone detoxifying enzyme, has been associated with Alzheimer’s disease (AD), suggesting that PON1 plays an important protective role in the brain. To study the involvement of PON1 in the development of AD and to elucidate the mechanism involved, we generated a new mouse model of AD, the *Pon1*^−/−^xFAD mouse, and examined how Pon1 depletion affects mTOR signaling, autophagy, and amyloid beta (Aβ) accumulation. To elucidate the mechanism involved, we examined these processes in N2a-APP_swe_ cells. We found that Pon1 depletion significantly downregulated Phf8 and upregulated H4K20me1; mTOR, phospho-mTOR, and App were upregulated while autophagy markers Bcln1, Atg5, and Atg7 were downregulated at the protein and mRNA levels in the brains of *Pon1*^─/─^5xFAD vs. *Pon1*^+/+^5xFAD mice. Pon1 depletion in N2a-APP_swe_ cells by RNA interference led to downregulation of Phf8 and upregulation of mTOR due to increased H4K20me1-*mTOR* promoter binding. This led to autophagy downregulation and significantly increased APP and Aβ levels. Phf8 depletion by RNA interference or treatments with Hcy-thiolactone or *N*-Hcy-protein metabolites similarly increased Aβ levels in N2a-APP_swe_ cells. Taken together, our findings define a neuroprotective mechanism by which Pon1 prevents Aβ generation.

## 1. Introduction 

Paraoxonase 1 (PON1), named for its ability to hydrolyze and inactivate the organophosphate paraoxon, is synthesized exclusively in the liver, circulates in the blood as a component of high-density lipoproteins (HDL) [1], and is present in many organs, including the brain [2]. In addition to protecting from organophosphate toxicity [3], PON1 protects against atherosclerosis induced by a high-fat diet [4] or ApoE depletion [5] in mice. Large-scale human studies showed that high arylesterase activity of PON1 protects from cardiovascular disease (CVD) in patients with coronary artery disease undergoing elective diagnostic coronary angiography [6,7] and in patients with chronic kidney disease [8], while low homocysteine thiolactonase activity of PON1 was associated with worse long-term mortality [9]. In the PREVEND prospective study involving 6902 participants, PON1 activity predicted CVD events [10]. The cardioprotective function of PON1 can be due both to its antioxidative function [4,6,11] and the ability to detoxify homocysteine (Hcy)-thiolactone [12,13,14,15], thereby attenuating lipid peroxidation, oxidative protein modification, and protein *N*-homocysteinylation. 

PON1 has also been implicated in Alzheimer’s disease (AD) [16,17], which can be expected given that AD has a significant vascular component [18]. For example, PON1 activity is lower in AD and dementia patients compared with healthy controls [19,20,21,22] and correlates with the severity of AD-related cognitive decline [23]. In patients with mild cognitive impairment, PON1 activity predicted global cognition, verbal episodic memory, and attention/processing speed [24]. In mice, *ApoE*^−/−^*Pon1*^−/−^ animals, which have severe carotid atherosclerosis [5], showed AD markers and impaired vasculature in their brains at 14 months, although it was not clear whether brain pathology was caused by *ApoE*^−/−^, *Pon1*^─/─^, or both knockouts [25]. In a mouse model of AD (Tg2576), immunohistochemical fluorescence signals for Pon1 protein in various regions of the brain were found to surround Aβ plaques but could not be colocalized to any brain cell type [26]. 

Deletion of the *Pon1* gene in mice impairs the metabolic conversion of Hcy-thiolactone to Hcy, increases brain Hcy-thiolactone levels, and makes the animals overly sensitive to the neurotoxicity of Hcy-thiolactone injections [12]. Studies of *Pon1*^−/−^ mouse brain proteome demonstrated that Pon1 interacts with diverse cellular processes, such as energy metabolism, anti-oxidative defenses, cell cycle, cytoskeleton dynamics, and synaptic plasticity, that are essential for brain homeostasis [27]. Clusterin (CLU or APOJ), involved in the transport of amyloid beta (Aβ) from plasma to brain in humans (reviewed in [28]), is carried on a distinct HDL subspecies that contains three major proteins: PON1, CLU, and APOA1 [29]. Notably, levels of Clu (ApoJ) are significantly elevated in the plasma of *Pon1*^−/−^ vs. *Pon1*^+/+^ mice [30]. These findings suggest that Pon1 plays a key role in brain homeostasis, possibly protecting from Aβ accumulation.

The present work was undertaken to examine the effects of Pon1 depletion on Aβ levels in a novel model of AD, the *Pon1*^─/─^5xFAD mouse, generated in the present study and to elucidate the mechanism involved. Because dysregulated mTOR signaling and autophagy have been implicated in Aβ accumulation in Alzheimer’s disease [31,32], and H4K20me1 demethylation by PHF8 is important for maintaining homeostasis of mTOR signaling [33], we studied how these processes are affected by Pon1 depletion in the mouse neuroblastoma N2a-APPswe cells and *Pon1*^─/─^5xFAD mice. We also examined how changes in these processes affect the behavioral performance of *Pon1*^─/─^5xFAD mice. 

## 2. Materials and Methods

### 2.1. Mice

*Pon1*^−/−^ [4] mice (kindly provided by Diane M. Shih) and 5xFAD mice [34] (The Jackson Laboratory, Bar Harbor, Maine, USA) on the C57BL/6J background were housed and bred at the New Jersey Medical School Animal Facility. 5xFAD mice overexpress the K670N/M671L (Swedish), I716V (Florida), and V717I (London) mutations in human APP (695), and M146L and L286V mutations in human PS1 and accumulate high levels of Aβ42 beginning around 2 months of age [35] (https://www.alzforum.org/research-models/5xfad-b6sjl) (accessed 27 December 2022). The *Pon1*^−/−^ mice were crossed with 5xFAD animals to generate *Pon1*^−/−^5xFAD mice and their *Pon1*^+/+^5xFAD sibling controls. Mouse *Pon1* genotype was established by PCR of tail clips DNA using the Pon1 forward primer p1 (5′-TGGGCTGCAGGTCTCAGGACTGA-3′), Pon1 exon 1 reverse primer p2 (5′-ATAGGAAGACCGATGGTTCT-3′), and neomycin cassette reverse primer p3 (5′-TCCTCGTGCTTTACGGTATCG-3′) [4]. The Pon1 genotype was also confirmed by RT-qPCR assays, which did not detect any Pon1 mRNA in the brains of *Pon1*^−/−^5xFAD mice but showed robust expression of Pon1 mRNA in the brains of their *Pon1*^+/+^5xFAD siblings. 

The 5xFAD genotype was established using human APP and PS1 primers (hAPP forward 5′-AGAGTACCAACTTGCATGACTACG-3′ and reverse 5′-ATGCTGGATAACTGCCTTCTTATC-3′; hPS1 forward 5′-GCTTTTTCCAGCTCTCATTTACTC-3′ and reverse 5′-AAAATTGATGGAATGCTAATTGGT-3′). The mice were fed a standard rodent chow diet (LabDiet 5010, Purina Mills International, St. Louis, MO, USA). 

Water supplemented with 1% methionine was used to induce hyperhomocysteinemia [12,27]. The high Met diet increases plasma total Hcy levels 5.6- and 10.4-fold in *Pon1*^−/−^ (from 8.5 to 48 μM) and *Pon1*^+/+^ mice (from 7.4 to 77 μM) [27]. Animal procedures were approved by the Institutional Animal Care and Use Committee at the New Jersey Medical School.

### 2.2. Brain Protein Extraction

Mice were euthanized by CO_2_ inhalation; the brains were collected and frozen on dry ice. Frozen brains were pulverized with dry ice using a mortar and pestle and stored at −80 °C. Proteins were extracted from the pulverized brains (50 ± 5 mg; 30 ± 3 mg brain was used for Aβ analyses) using RIPA buffer (4 *v*/*w*, containing protease and phosphatase inhibitors) with sonication (Bandelin SONOPLUS HD 2070) on wet ice (three sets of five 1-s strokes with 1 min cooling interval between strokes). Brain extracts were clarified by centrifugation (15,000× *g*, 30 min, 4 °C) and clear supernatants containing 8–12 mg protein/mL were collected (RIPA-soluble fraction). Protein concentrations were measured with BCA kit (Thermo Fisher Scientific, Waltham, MA, USA).

For Aβ analyses, pellets remaining after protein extraction with RIPA buffer were re-extracted by brief sonication in 2% SDS, centrifuged (15,000× *g*, 15 min, room temperature), and the supernatants collected again (SDS-soluble fraction). The SDS-extracted pellets were then extracted by sonication in 70% formic acid (FA), centrifuged, and the supernatants were collected (the FA-soluble fraction) [35].

### 2.3. Aβ Quantification

Aβ was quantified using a dot blot assay [36]. Briefly, brain protein extracts (1 µL) were spotted onto the nitrocellulose membranes and dried (37 °C, 1 h). The membranes were washed with TBST buffer (RT, 15 min) and blocked with 5% BSA in TBST buffer (RT, 1 h). Blocked membranes were washed three times with TBST buffer (10 min each) and incubated with monoclonal anti-Aβ antibody (CS #8243; 4 °C, 16 h). Membranes were then washed three times with TBST buffer (10 min each) and incubated with goat horseradish peroxidase-conjugated anti-rabbit IgG secondary antibody. Positive signals were detected using Western Bright Quantum-Advansta K12042-D20 and GeneGnome XRQ NPC chemiluminescence detection system. Signal intensity was assessed using the Gene Tools program from Syngene.

### 2.4. Cell Culture and Treatments

Mouse neuroblastoma N2a-APPswe cells, harboring a human APP transgene with the K670N and M671L Swedish mutations [37] were grown (37 °C, 5% CO_2_) in DMEM/F12 medium (Thermo Fisher Scientific, Waltham, MA, USA) supplemented with 5% FBS, non-essential amino acids, and antibiotics (MilliporeSigma, Saint Louis, MO, USA). 

After cells reached 70–80% confluency, the monolayers were washed twice with PBS and overlaid with DMEM medium without methionine (Thermo Scientific), supplemented with 5% dialyzed fetal bovine serum (FBS) (MilliporeSigma) and non-essential amino acids. L-Hcy-thiolactone (20 and 200 μM) (MilliporeSigma) or *N*-Hcy-protein (10 and 20 μM), prepared as described in ref. [38], were added, and the cultures were incubated at 37 °C in a 5% CO_2_ atmosphere for 24 h.

For gene silencing, siRNAs targeting the *Pon1* (Cat. # s71950 and s71951) or *Phf8* gene (Cat. # s115808, and s115809) (Thermo Scientific) were transfected into cells maintained in Opti-MEM medium by 48-h Lipofectamine RNAiMax (Thermo Scientific) treatments. Cellular RNA for RT-qPCR analysis was isolated as described in Section 2.5. For protein extraction, RIPA buffer (MilliporeSigma) was used according to the manufacturer’s protocol. 

### 2.5. Western Blots 

Proteins were separated by SDS-PAGE on 10% gels (20 µg protein/lane) and transferred to a PVDF membrane (Bio-Rad) for 20 min at 0.1 A, 25 V using the Trans Blot Turbo Transfer System (Bio-Rad). After blocking with 5% bovine serum albumin in TBST buffer (1 h, room temperature), the membranes were incubated with monoclonal anti-Pon1 (ab126597, Abcam, Cambridge, MA, USA), anti-Phf8 (Abcam, ab36068), anti-H4K20me1 (Abcam ab177188), anti-mTOR (Cell Signaling Technology, Davnvers, MA, USA, CS #2983), anti-pmTOR Ser2448 (CS, #5536), anti-Atg5 (CS, #12994), anti-Atg7 (CS, #8558), anti-Beclin-1 (CS, #3495), anti-Gapdh (CS, #5174), or anti-App (Abcam, ab126732) overnight at 4 °C. Membranes were washed three times with TBST buffer, for 10 min each, and incubated with goat anti-rabbit IgG secondary antibody conjugated with horseradish peroxidase. Positive signals were detected using Western Bright Quantum-Advansta K12042-D20 and GeneGnome XRQ NPC chemiluminescence detection system. Band intensity was calculated using the Gene Tools program from Syngene.

### 2.6. RNA Isolation, cDNA Synthesis, RT-qPCR Analysis

Total RNA was isolated using Trizol reagent (MilliporeSigma). cDNA synthesis was conducted using Revert Aid First cDNA Synthesis Kit (Thermo Fisher Scientific) according to the manufacturer’s protocol. Nucleic acid concentration was measured using NanoDrop (Thermo Fisher Scientific). RT-qPCR was performed with SYBR Green Mix and CFX96 thermocycler (Bio-Rad, Hercules, CA, USA). The 2^(−ΔΔCt)^ method was used to calculate the relative expression levels [39]. Data analysis was performed with the CFX Manager™ Software, Microsoft Excel, and Statistica. RT-qPCR primer sequences are listed in Appendix A.

### 2.7. Chromatin Immunoprecipitation Assay

For CHIP assays we used CUT&RUN Assay Kit #86652 (Cell Signaling Technology, Danvers, MA, USA) following the manufacturer’s protocol. Each ChIP assay was repeated three times. Briefly, for each reaction, we used 100,000 cells. Cells were trypsinized and harvested, washed 3× in ice-cold PBS, and bound to concanavalin A-coated magnetic beads for 5 min, at RT. Cells were then incubated (4 h, 4 °C) with 2.5 µg of anti-PHF8 antibody (Abcam, ab36068) or anti-H4K20me1 antibody (Abcam, ab177188) in the antibody-binding buffer plus digitonin that permeabilizes cells. Next, cells were treated with pAG-MNase (1 h, 4 °C), washed, and treated with CaCl_2_ to activate DNA digestion (0.5 h, 4°C). Cells were then treated with the stop buffer and spike-in DNA was added for each reaction for signal normalization, and incubated (10–30 min, 37 °C). Released DNA fragments were purified using DNA Purification Buffers and Spin Columns (CS #14209) and quantified by RT-qPCR using primers targeting the promoter, upstream, and downstream regions of the *mTOR* gene (Appendix A). Rabbit (DA1E) mAb IgG XP^®^ Isotype Control included in the CUT&RUN kit did not afford any signals in the RT-qPCR assays targeting *mTOR*. 

### 2.8. Confocal Microscopy, Aβ Quantification in N2a-APPswe Cells

Mouse neuroblastoma N2a-APPswe cells were cultured in Millicell EZ SLIDE 8-well glass slides (Merck, Darmstadt, Germany). After treatments, cells were washed 3 times with PBS for 10 min. Cells were fixed with 4% PFA (MilliporeSigma) (37 °C, 15 min), washed 3 times with PBS buffer, permeabilized with 0.1% Triton X-100 solution (RT, 20 min), blocked with 0.1% BSA (RT, 1h), and incubated with anti-Aβ antibody (CS #8243; 4 °C, 16 h). Cells were then washed 3 times with PBS and stained with secondary antibody Goat Anti-Rabbit IgG H&L (Alexa Fluor^®^ 488) (Abcam, ab150077; RT, 1 h) to visualize and quantify Aβ. DAPI (Vector Laboratories, Newark, CA, USA) was used to visualize nuclei. Fluorescence signals were monitored by using a Zeiss LSM 880 confocal microscope with a 488 nm filter for the Alexa Fluor^®^ 488 (Aβ) and 420–480 nm filter for DAPI, taking a *z* stack of 20–30 sections with an interval of 0.66 μm and a range of 15 μm. Zeiss Plan-Apochromat X40/1.2 Oil differential interference contrast objective were used for imaging. Images were quantified with the ImageJ Fiji 2.9.0 software (NIH, Bethesda, MD, USA).

### 2.9. Behavioral Testing

#### 2.9.1. Hindlimb Test

The hindlimb clasping test is used to assess neurodegeneration in mouse models [40]. For this test, mice were suspended by the base of the tail and videotaped for 10 s. Three separate trials were taken over three consecutive days. Hindlimb clasping was scored from 0 to 3: 0 = hindlimbs splayed outward and away from the abdomen; 1 = one hindlimb retracted inwards towards the abdomen for at least 50% of the observation period; 2 = both hindlimbs partially retracted inwards towards the abdomen for at least 50% of the observation period; and 3 = both hindlimbs completely retracted inwards towards the abdomen for at least 50% of the observation period. Hindlimb clasping scores were added together for the three separate trials. 

#### 2.9.2. Ledge Test

The ledge test is used to assess motor deficits in rodent models of CNS disorders [41]. Typically, mice walk along the ledge of a cage and try to descend back into the cage. Three separate trials were taken for each mouse. The ledge test was scored from 0 to 3 points: 0 = a mouse walked along the ledge without slipping and lowered itself back into the cage using paws; 1 = the mouse lost its footing during walking along the ledge but otherwise appeared coordinated; 2 = the mouse did not effectively use its hind legs and landed on its head rather than paws when descending into the cage; and 3 = the mouse fell of the ledge or was shaking and/or barely moving.

#### 2.9.3. Cylinder Test

The cylinder test is used to assess sensorimotor function in rodent models of CNS disorders. A mouse is placed in a transparent 500 mL plastic cylinder. The number of times the mouse rears up and touches the cylinder wall during a period of 3 min is counted. A rear is defined as a vertical movement with both forelimbs off the floor so that the mouse is standing only on its hindlimbs. At the end of 3 min, the mouse was removed and placed back into its home cage. Because spontaneous activity in the cylinder is affected by repeated testing, resulting in reduced activity over time, mice were tested only once in their lifetime.

### 2.10. Statistical Analysis

The results were calculated as mean ± standard deviation. A two-sided unpaired t test was used for comparisons between two groups of variables; *p* < 0.05 was considered significant. Statistical analysis was performed using Statistica, Version 13 (TIBCO Software Inc., Palo Alto, CA, USA, http://statistica.io) (accessed 2 November 2022).

## 3. Results

### 3.1. Pon1 Depletion Downregulates the Expression of Histone Demethylase Phf8 and Increases the H4K20me1 Epigenetic Mark in Mouse Brain

To determine if Pon1 interacts with Phf8, we quantified Phf8 protein in the brains of *Pon1*^−/−^5xFAD mice and their *Pon1*^+/+^5xFAD sibling controls by Western blotting. We also examined the effects of hyperhomocysteinemia (HHcy), induced by providing 1% methionine in drinking water, on the Pon1–Phf8 interaction. Pictures of Western blots are shown in Figure 1I and Figure 2B, while quantification of individual proteins is illustrated by corresponding bar graphs in Figure 1 and Figure 2 for 5-month-old and 12-month-old mice, respectively. We found that Phf8 protein was significantly downregulated in the brains of *Pon1*^−/−^5xFAD mice vs. *Pon1*^+/+^5xFAD sibling controls in animals fed with a standard chow diet (5-month-old: from 1.0 ± 0.1 to 0.68 ± 0.15, *P*_genotype_ = 2 × 10^−5^, Figure 1A; 12-month-old: from 1.0 ± 0.2 to 0.65 ± 0.12, *P*_genotype_ = 1 × 10^−4^, Figure 2A). Reduced expression of Phf8 in *Pon1*^−/−^5xFAD vs. *Pon1*^+/+^5xFAD brains was also observed in mice fed with the HHcy diet (from 0.76 ± 0.11 to 0.60 ± 0.10, *P*_genotype_ = 0.001; Figure 1A). 

HHcy diet significantly downregulated Phf8 expression in the brains of *Pon1*^+/+^5xFAD mice (to 0.76 ± 0.11, *P*_diet_ = 6 × 10^−5^). In contrast, Phf8 expression in the brains of *Pon1*^−/−^5xFAD mice was essentially not affected by the HHcy diet (0.60 ± 0.19 vs. 0.68 ± 0.15, *P*_diet_ = 0.099) (Figure 1A).

The histone H4K20me1 epigenetic mark was significantly upregulated in *Pon1*^−/−^5xFAD vs. *Pon1*^+/+^5xFAD brains (5-month-old: 1.74-fold, *P*_genotype_ = 1 × 10^−7^, Figure 1B; 12-month-old: 1.41-fold, *P*_genotype_ = 1 × 10^−4^, Figure 2). Upregulated expression of H4K20me1 in 5-month-old *Pon1*^−/−^5xFAD vs. *Pon1*^+/+^5xFAD brains was also observed in mice fed with the HHcy diet (from 1.58 ± 0.27 to 1.87 ± 0.24, *P*_genotype_ = 0.030; Figure 1B). 

HHcy diet significantly upregulated H4K20me1 levels in 5-month-old *Pon1*^+/+^ mice (1.6-fold, *P*_diet_ = 6 × 10^−6^) but not in *Pon1*^─/─^ animals (1.74- vs. 1.87-fold, *P*_diet_ = 0.275; Figure 1B).

### 3.2. Pon1 Depletion Upregulates mTOR and Inhibits Autophagy in Mouse Brain

Because Phf8/H4K20me1 regulate mTOR signaling, we next examined the effects of Pon1 depletion on levels of mTOR and its active form, phosphorylated at Ser2448 (pmTOR). We found that mTOR protein was significantly upregulated in the brains of *Pon1*^─/─^5xFAD vs. *Pon1*^+/+^5xFAD mice (5-month-old: 1.69-fold, *P*_genotype_ = 2 × 10^−10^, Figure 1C; 12-month-old: 1.39-fold, *P*_genotype_ = 4 × 10^−5^, Figure 2A). Upregulated expression of mTOR in *Pon1*^─/─^5xFAD vs. *Pon1*^+/+^5xFAD brains was also observed in mice fed with the HHcy diet (from 1.43 ± 0.18 to 1.97 ± 0.19, *P*_genotype_ = 2 × 10^−5^; Figure 1C).

HHcy diet significantly upregulated mTOR protein expression in *Pon1*^─/─^5xFAD mice (1.97 ± 0.19 vs. 1.69 ± 0.12, *P*_diet_ = 0.003) and *Pon1*^+/+^5xFAD animals (1.43 ± 0.18 vs. 1.00 ± 0.09, *P*_diet_ = 5 × 10^−6^) (Figure 1C). 

Because mTOR is activated by phosphorylation, we quantified mTOR phosphorylated at Ser2448 (pmTOR). We found that pmTOR was also significantly upregulated in the brains of *Pon1*^─/─^5xFAD vs. *Pon1*^+/+^5xFAD mice (5-month-old: 1.69-fold, *P*_genotype_ = 2 × 10^−10^, Figure 1C; 12-month-old: 1.86-fold, *P*_genotype_ = 3 × 10^−8^, Figure 2A). Upregulated expression of pmTOR in *Pon1*^─/─^5xFAD vs. *Pon1*^+/+^5xFAD brains was also observed in mice fed with the HHcy diet (1.95 ± 0.17 vs. 1.56 ± 0.26, *P*_genotype_ = 0.002) (Figure 1D). 

HHcy diet significantly elevated pmTOR levels in *Pon1*^+/+^5xFAD mice (1.56 ± 0.26 vs. 1.00 ± 0.18, *P*_diet_ = 4 × 10^−5^) but not in *Pon1*^−/−^5xFAD mice (1.95 ± 0.17 vs. 1.87 ± 0.30, *P*_diet_ = 0.528 (Figure 1D). 

Overall, the effects of the *Pon1*^−/−^ genotype on mTOR and pmTOR levels were attenuated by the HHcy diet (Figure 1C,D). These findings indicate that Pon1 depletion upregulated pmTOR to a similar extent as mTOR, suggesting that the *Pon1*^─/─^ genotype affects mTOR signaling mostly at the level of mTOR protein expression. 

Because mTOR is a major regulator of autophagy, we quantified autophagy-related proteins in *Pon1*^─/─^5xFAD mice. We found that the regulators of autophagosome assembly, Bcln1, Atg5, and Atg7, were significantly downregulated in the brains of *Pon1*^─/─^5xFAD vs. *Pon1*^+/+^5xFAD sibling controls (by 22–35%, *P*_genotype_ = 1 × 10^−7^ to 1 × 10^−4^, Figure 1E–G; 12-month-old: by 24–37%, *P*_genotype_ = 2 × 10^−5^ to 3 × 10^−4^, Figure 2A). The HHcy diet significantly decreased Bcln1, Atg5, and Atg7 expression in 5-month-old *Pon1*^+/+^5xFAD mice (by 23–28%, *P*_diet_ = 2 × 10^−5^ to 2 × 10^−9^). In 5-month-old *Pon1*^─/─^5xFAD mice, the HHcy diet also significantly decreased Bcln1 (0.68 vs. 0.80, *P*_diet_ = 0.003) and Atg5 levels (0.66 vs. 0.74, *P*_diet_ = 0.008); however, Atg7 levels were essentially not affected by the HHcy diet in *Pon1*^─/─^5xFAD mice (0.63 ± 0.05 vs. 0.65 ± 0.05, *P*_diet_ = 0.714). Overall, the effects of the *Pon1*^−/−^ genotype on the brain Bcln1, Atg5, and Atg7 levels were attenuated by the HHcy diet (Figure 1E–G). These findings indicate that autophagy was impaired by the *Pon1*^─/─^genotype.

### 3.3. Pon1 Depletion Upregulates APP Protein Expression in Mouse Brain

We found that APP protein was significantly elevated in the brains of *Pon1*^−/−^5xFAD mice vs. *Pon1*^+/+^5xFAD sibling controls in mice fed with a standard diet (5-month-old: 1.42-fold, *P*_genotype_ = 2 × 10^−8^; Figure 1H; 12-month-old: 1.39-fold, *P*_genotype_ = 3 × 10^−6^, Figure 2A). Upregulated expression of APP protein in 5-month-old *Pon1*^−/−^5xFAD vs. *Pon1*^+/+^5xFAD brains was also observed in mice fed with the HHcy diet (from 1.76 ± 0.08 to 1.92 ± 0.10, *P*_genotype_ = 0.005; Figure 1H). 

Met diet increased APP protein levels in the brains of 5-month-old *Pon1*^+/+^5xFAD mice (1.76-fold, *P*_diet_ = 1 × 10^−13^) and, to a lesser extent, in *Pon1*^─/─^5xFAD animals (1.35-fold, from 1.42 to 1.92, *P*_diet_ = 4 × 10^−7^) (Figure 1H).

### 3.4. Pon1 Gene Exerts Transcriptional Control on the Expression of Phf8, mTOR, Autophagy-Related Proteins, and APP in Pon1^─/─^5xFAD Mice

To determine if the observed changes in the protein levels of Phf8, mTOR, autophagy-related proteins, and APP are caused by the transcriptional effects of the *Pon1*^─/─^ genotype, we quantified the corresponding mRNAs by RT-qPCR. We found that Phf8 mRNA was significantly downregulated in the brains of *Pon1*^−/−^5xFAD mice vs. *Pon1*^+/+^5xFAD sibling controls in animals fed with a standard chow diet (5-month-old: from 1.00 ± 0.15 to 0.66 ± 0.09, *P*_genotype_ = 1 × 10^−4^, Appendix A; 12-month-old: from 1.00 ± 0.16 to 0.76 ± 0.13, *P*_genotype_ = 4 × 10^−4^, Figure 2C). HHcy did not affect the effects of the Pon1 genotype on Phf8 mRNA: reduced expression of Phf8 in the brains of 5-month-old *Pon1*^−/−^5xFAD vs. *Pon1*^+/+^5xFAD mice was observed in mice fed with the Met diet (from 0.63 ± 0.27 to 0.37 ± 0.23, *P*_genotype_ = 0.048; Appendix A).

HHcy significantly downregulated Phf8 mRNA expression in the brains of 5-month-old 5xFAD mice, regardless of *Pon1* genotype: from 1.00 ± 0.15 in mice fed with a standard diet to 0.63 ± 0.27 in animals fed with the Met diet, *P*_diet_ = 0.002 in *Pon1*^+/+^5xFAD mice, and from 0.66 ± 0.09 (std diet) to 0.37 ± 0.23 (Met diet), *P*_diet_ = 0.002 in *Pon1*^─/─^5xFAD animals (Appendix A).

We found that mTOR mRNA was significantly upregulated in the brains of *Pon1*^─/─^5xFAD vs. *Pon1*^+/+^5xFAD mice (5-month-old: 1.55-fold, *P*_genotype_ = 0.006, Appendix A; 12-month-old: 1.32-fold, *P*_genotype_ = 4 × 10^−5^, Figure 2C). However, HHcy abrogated the effects of the *Pon1* genotype on mTOR mRNA expression: similar levels of mTOR mRNA were found in *Pon1*^−/−^5xFAD and *Pon1*^+/+^5xFAD mice fed with the high Met diet (1.79 ± 0.55 and 1.46 ± 0.61, respectively, *P*_genotype_ = 0.258; Appendix A).

HHcy diet significantly upregulated mTOR mRNA in *Pon1*^+/+^5xFAD mice (1.46 ± 0.61 vs. 1.00 ± 0.15, *P*_diet_ = 0.044) but not in *Pon1*^─/─^5xFAD animals (1.79 ± 0.55 vs. 1.55 ± 0.46, *P*_diet_ = 0.352) (Appendix A).

We also found that mRNA for the regulators of autophagosome assembly, Bcln1, Atg5, and Atg7, were downregulated in the brains of *Pon1*^─/─^5xFAD vs. *Pon1*^+/+^5xFAD sibling controls (Bcln1 and Atg7 mRNA by 31% and 22%, *P*_genotype_ = 0.005 and 0.008, respectively, Appendix A; 12-month-old: by 13–36%, *P*_genotype_ = 2 × 10^−5^ to 0.040, Figure 2C). Met diet significantly decreased Bcln1 and Atg7 mRNA expression in 5-month-old *Pon1*^+/+^5xFAD mice (by 18–20%, *P*_diet_ = 0.044 and 0.008, respectively) but not in *Pon1*^─/─^5xFAD animals. The Atg5 mRNA level was not affected by the Met diet regardless of *Pon1* genotype. However, Atg5 mRNA was significantly reduced by the *Pon1*^─/─^ genotype in mice fed with the Met diet but in *Pon1*^−/−^5xFAD mice. Overall, the effects of the *Pon1*^─/─^ genotype on the brain Bcln1 and Atg7 levels were attenuated by the HHcy diet (Appendix A). 

We found that APP mRNA was significantly elevated in the brains of *Pon1*^─/─^5xFAD mice vs. *Pon1*^+/+^5xFAD sibling controls in mice fed with a standard diet (5-month-old: 1.52-fold, *P*_genotype_ = 0.002; Appendix A; 12-month-old: 1.59-fold, *P*_genotype_ = 0.003, Figure 2C). Upregulated expression of APP mRNA in 5-month-old *Pon1*^─/─^5xFAD vs. *Pon1*^+/+^5xFAD brains was also observed in mice fed with the HHcy diet (from 1.76 ± 0.08 to 1.92 ± 0.10, *P*_genotype_ = 0.005; Appendix A).

Met diet increased APP mRNA levels in the brains of 5-month-old *Pon1*^+/+^5xFAD mice (1.75-fold, *P*_diet_ = 0.010) but not in *Pon1*^─/─^5xFAD animals (*P*_diet_ = 0.482) and abrogated the effects of the *Pon1*^─/─^ genotype on APP mRNA (Appendix A). As expected, Pon1 mRNA was absent in *Pon1*^─/─^5xFAD brains (Appendix A). Met diet did not affect Pon1 mRNA in *Pon1*^+/+^5xFAD mice brains (Appendix A). 

These findings indicate that the *Pon1* gene exerts transcriptional control over the expression of Phf8, mTOR, autophagy-related proteins, and APP in the mouse brain. 

### 3.5. Pon1 Gene Silencing Downregulates the Histone Demethylase Phf8, Upregulates H4K20me1 Epigenetic Mark, mTOR and pmTOR, APP, and Inhibits Autophagy in Mouse Neuroblastoma N2a-APPswe Cells

To elucidate the mechanism by which Pon1 depletion impacts Phf8 and its downstream effects on mTOR, autophagy, and APP, we first examined whether the findings in *Pon1*^−/−^ mice can be recapitulated in cultured mouse neuroblastoma N2a-APPswe cells that overproduce Aβ from a mutated human APP transgene [38]. We silenced the *Pon1* gene in these cells by RNA interference using *Pon1*-targeting siRNA and studied how the silencing impacts Phf8 and its downstream effects. Changes in specific protein levels in *Pon1*-silenced and control cells were analyzed by Western blotting using Gapdh protein as a reference. 

We found that the Pon1 protein level was reduced by 71% in *Pon1*-silenced cells (*p* = 1 × 10^─5^; Figure 3A). We also found that the histone demethylase Phf8 protein level was significantly downregulated (by 33%, *p* = 4 × 10^─4^; Figure 3B), while the histone H4K20me1 level was significantly upregulated (1.70–1.76-fold, *p* = 0.001; Figure 3C) in *Pon1*-silenced N2a-APPswe cells. 

At the same time, the mTOR protein was significantly upregulated in *Pon1*-silenced N2a-APPswe cells (1.7-fold, *p* = 0.001; Figure 3D), as were pmTOR (1.6-fold, *p* = 2 × 10^−5^; Figure 3E) and APP (1.6-fold, *p* = 1 × 10^−4^; Figure 3I), while autophagy-related proteins Bcln1, Atg5, and Atg7 (Figure 3F–H, respectively) were significantly downregulated (by 33–45%, *p* = 2 × 10^−4^ to 0.003).

The Western blot results show that the changes in Phf8, H4K20m31, mTOR signaling, autophagy, and APP induced by *Pon1* gene silencing in N2a-APPswe cells (Figure 3) recapitulate the in vivo findings in the *Pon1*^−/−^5xFAD mouse brain (Figure 1 and Figure 2).

### 3.6. Pon1 Gene Silencing Increases H4K20me1 Biding to mTOR Promoter in N2a-APPswe Cells

To determine whether increased levels of the histone H4K20me1 mark can promote mTOR gene expression by binding to its promoter in Pon1-depleted cells, we carried out ChIP experiments using anti-H4K20me1 antibody (Figure 4). The *Pon1* gene was silenced by transfecting N2a-APPswe cells using two different *Pon1*-targeting siRNAs. The cells were permeabilized and treated with anti-H4K20me1 antibody and a recombinant micrococcal nuclease-protein A/G. DNA fragments released form N2a-APPswe cells were quantified by RT-qPCR using primers targeting the transcription start site (TSS) of the mTOR gene as well as upstream (UP) and downstream (DOWN) regions. 

We found that in *Pon1*-silenced N2a-APPswe cells, the binding of H4K20me1 was significantly increased at the mTOR TSS (1.8 to 2.3-fold, *p* = 4 × 10^−5^), mTOR UP (2.0 to 2.2-fold, *p* = 2 × 10^−5^), and mTOR DOWN sites (1.4 to 1.6-fold, *p* = 1 × 10^−4^) (Figure 4A). Importantly, in *Pon1*-silenced cells there were significantly more DNA fragments from the mTOR TSS (2.3 ± 0.2 and 1.8 ± 0.2 for siRNA *Pon1* #1 and #2, respectively) than from the DOWN site (1.4 ± 0.2 and 1.6 ± 0.1 for siRNA *Pon1* #1 and #2, respectively; *p* = 0.004). There were also more DNA fragments from the UP site than from the DOWN site (2.2 ± 0.2 and 2.1 ± 0.2 for siRNA *Pon1* #1 and #2 vs. 1.4 ± 0.2 and 1.6 ± 0.1 for siRNA *Pon1* #1 and #2; *p* = 0.0004). Numbers of DNA fragments from the TSS and UP sites were similar (*p* = 0.713) (Figure 4A). Control experiments showed that the binding of H3K4me3 to RPL30 intron was not affected by *Pon1* gene silencing (Figure 4B). These findings indicate that *Pon1* gene silencing induces H4K20me1 binding at the *mTOR* gene, significantly higher at the mTOR TSS and UP site than at the DOWN site in *Pon1*-silenced cells.

CHIP experiments using anti-Phf8 antibody showed that *Pon1* gene silencing or treatments with Hcy-thiolactone or *N*-Hcy-protein did not affect binding of Phf8 to the mTOR gene.

### 3.7. Pon1 Depletion Increases Aβ Accumulation in N2a-APPswe Cells

To determine whether Pon1 depletion affects Aβ accumulation, we silenced the *Pon1* gene by RNA interference and quantified Aβ in N2a-APPswe cells by fluorescence confocal microscopy using anti-Aβ antibody. The *Pon1* gene was silenced by transfection with two different siRNAs targeting *Pon1*; the cells were permeabilized, treated with anti-Aβ antibody, and Aβ was visualized with fluorescent secondary antibody and quantified. Representative confocal microscopy images are shown in Figure 5A. We found that *Pon1* gene silencing led to increased Aβ generation manifested by significantly increased area (from 173 ± 27 and 162 ± 22 μm^2^ for -siRNA and siRNAscr controls, respectively, to 225 ± 28 μm^2^ for siRNA Pon1 #1, *p* = 0.013) and average size (from 0.63 ± 0.02 and 0.61 ± 0.06 for -siRNA and siRNAscr controls, respectively, to 1.29 ± 0.17 μm^2^ and 0.99 ± 0.09 μm^2^ for siRNA Pon1 #1 and #2, respectively; *p* = 1 × 10^−4^) of fluorescent Aβ puncta in Pon1 siRNA-treated N2a-APPswe cells compared with siRNAscr or -siRNA controls (Figure 5B). Signal intensity increased from 1.00 ± 0.16 and 0.86 ± 0.28 for -siRNA and siRNAscr controls, respectively, to 2.08 ± 0.27 and 2.01 ± 0.23 for siRNA Pon1 #1 and #2, respectively; *p* = 3 × 10^−4^) (Figure 5B).

Because Pon1 depletion elevates Hcy-thiolactone and *N*-Hcy-protein in mice [12], we examined whether any of these metabolites can induce Aβ accumulation in N2a-APPswe cells. In cells treated with Hcy-thiolactone (20–200 μM) or *N*-Hcy-protein (10–20 μM), there was significantly more Aβ, manifested by significantly increased area of fluorescent Aβ puncta in confocal immunofluorescence images compared with control-siRNA and siRNAscr (Figure 5C,D). However, while treatments with Hcy-thiolactone led to increased size and signal intensity of the fluorescent Aβ puncta, treatments with *N*-Hcy-protein did not (Figure 5D), suggesting different effects of Hcy-thiolactone and *N*-Hcy-protein on the structure of Aβ deposits. These findings suggest that Hcy-thiolactone and *N*-Hcy-protein contribute to elevated Aβ levels induced by *Pon1* gene silencing. 

### 3.8. Pon1 Depletion Increases Aβ Accumulation in Pon1^−/−^5xFAD Mice

Aβ was extracted from brains of 5- and 12-month-old mice fed with a standard chow diet, and from 5-month-old mice with the HHcy diet (1% Met in drinking water) since weaning at the age of 1 month. SDS-soluble and formic acid (FA)-soluble Aβ fractions, which contain the bulk of Aβ [36], as well as a minor RIPA-soluble Aβ fraction were obtained. Aβ was quantified in these fractions by a dot blot assay with a monoclonal anti-Aβ antibody [37].

We found that RIPA- and SDS-soluble Aβ was significantly elevated (*P*_genotype_ = 2 × 10^−5^ and 1 × 10^−8^, respectively), and FA-soluble Aβ tended to be elevated (*P*_genotype_ = 0.058) in the brains of 12-month-old *Pon1*^−/−^5xFAD mice vs. *Pon1*^+/+^5xFAD sibling controls fed with a standard diet (Figure 6A). Similarly, elevated Aβ was found in 5-month-old *Pon1*^−/−^5xFAD vs. *Pon1*^+/+^5xFAD mice fed with a standard diet (Figure 6B) or the HHcy diet (Figure 6C). This indicates that neither age nor HHcy influenced the effects of the *Pon1*^−/−^ genotype on Aβ levels. 

However, the HHcy diet significantly elevated RIPA-, SDS-, and FA-soluble Aβ in 5-month-old *Pon1*^−/−^5xFAD mice (from 1.21 to 1.97, *P*_diet_ = 1 × 10^−4^; 2.62 to 3.09, *P*_diet_ = 0.034; 1.88 to 3.33, *P*_diet_ = 1 × 10^−6^, respectively) and in 5-month-old *Pon*1^+/+^5xFAD mice, (from 1.00 to 1.95, *P*_diet_ = 0.002; 1.00 to 1.91, *P*_diet_ = 4 × 10^−4^; 1.00 to 1.45, *P*_diet_ = 5 × 10^−4^, respectively) (Figure 6C). This indicates that HHcy and *Pon1*^−/−^ genotype exert similar effects on Aβ levels.

### 3.9. Pon1 Depletion Does Not Induce Sensorimotor Deficits

To examine the effects of Pon1 depletion on neurodegeneration and sensorimotor activity, 12-month-old *Pon1*^−/−^5xFAD mice and their *Pon1*^+/+^5xFAD sibling controls were assessed in the hindlimb clasping, ledge, and cylinder tests. 

The hindlimb test showed a similar degree of clasping (scores) in *Pon1*^−/−^5xFAD mice vs. their *Pon1*^+/+^5xFAD littermates (2.24 ± 0.44 vs. 2.08 ± 0.43, *p* = 0.335; Appendix A). These findings indicate that the *Pon1*^−/−^ genotype did not induce neurodegeneration in *Pon1*^−/−^5xFAD mice relative to *Pon1*^+/+^5xFAD animals. 

The ledge test showed similar performances (scores) in *Pon1*^−/−^5xFAD mice vs. their *Pon1*^+/+^5xFAD littermates (2.07 ± 0.43 vs. 1.97 ± 0.38, *p* = 0.589; Appendix A). The cylinder test also showed similar performances (number of rears) in *Pon1*^−/−^5xFAD mice vs. their *Pon1*^+/+^5xFAD littermates (8.5 ± 6.0 vs. 10.4 ± 5.1, *p* = 0.307; Appendix A). These findings indicate that the *Pon1*^−/−^ genotype did not induce sensorimotor deficits in *Pon1*^−/−^5xFAD mice relative to *Pon1*^+/+^5xFAD animals. 

## 4. Discussion

In previous studies, we found that Pon1 is a Hcy-thiolactone-hydrolyzing enzyme [13] and that Pon1 depletion in mice elevated brain Hcy-thiolactone and *N*-Hcy-protein [12], increased the animals’ susceptibility to Hcy-thiolactone-induced seizures [12], and resulted in pro-neurodegenerative changes in brain proteome [27], suggesting that Pon1 plays an important protective role in brain homeostasis.

Our present findings show that Pon1 protects from amyloidogenic APP processing to Aβ in mice brains (Figure 6) and unravel the mechanistic basis of the protective role of Pon1 in the CNS. Specifically, we found that Pon1 depletion downregulated histone demethylase Phf8 both at the protein and mRNA level, increased H4K20me1 binding at the mTOR promotor (Figure 4A), and upregulated mTOR expression and phosphorylation in the mouse brain (Figure 1C,D) and neuroblastoma N2a-APPswe cells (Figure 3D,E). Treatments with Hcy-thiolactone and *N*-Hcy-protein, metabolites that are elevated in *Pon1*^−/−^ mice, also increased H4K20me1 binding at the mTOR promotor in N2a-APPswe cells (Figure 4C). This suggests that Pon1 is a negative regulator of mTOR signaling by controlling levels of Hcy metabolites that affect binding of H4K20me1 at the mTOR promotor. The effects of Hcy-thiolactone and *N*-Hcy-protein on mTOR are explained by findings that Phf8, the regulator of mTOR expression, was downregulated by Pon1 depletion (Figure 1A and Figure 2A), whereas H4K20me1 was upregulated (Figure 1B and Figure 2A). These findings provide direct mechanistic evidence linking Hcy-thiolactone and *N*-Hcy-protein with dysregulated mTOR signaling and its downstream consequences such as downregulation of autophagy and upregulation of Aβ. This mechanism is further supported by our findings that Phf8 depletion by RNA interference affected mTOR, autophagy, APP, and Aβ, similar to treatments with Hcy-thiolactone or *N*-Hcy-protein [42]

In the present study, we found that depletion of Pon1 upregulated APP in the *Pon1*^−/−^5xFAD mouse brain (Figure 1H, Figure 2 and Appendix A) and in mouse neuroblastoma N2a-APPswe cells (Figure 3I). In contrast, depletion of Phf8 did not affect APP expression [42]. These findings suggest that Pon1 interacts with APP in the *Pon1*^−/−^5xFAD mouse brain while Phf8 does not. However, whether the Pon1–APP interaction is direct or indirect remains to be determined.

Although Pon1 depletion in mouse neuroblastoma N2a-APPswe cells downregulated Phf8 (Figure 1A and Figure 2A) and upregulated APP (Figure 1H and Figure 2A) and Aβ (Figure 5), depletion of Phf8 upregulated Aβ but not APP [42]. These findings suggest that two pathways can lead to increased Aβ generation in Pon1-depleted brains and neural cells. One pathway involves Hcy metabolites, which upregulate APP, while another pathway involves impaired Aβ clearance due to downregulated autophagy.

Notably, Pon1 depletion caused changes in the Phf8- > H4K20me1- > mTOR- > autophagy pathway akin to the changes induced by HHcy in the mouse brain (Figure 1) and neuroblastoma cells (Figure 3). Pon1 depletion or HHcy similarly increased accumulation of Aβ in the brain (Figure 6). Our previous work showed that a common primary biochemical outcome of Pon1 depletion or of HHcy was essentially the same: HHcy caused elevation of Hcy-thiolactone and *N*-Hcy-protein [43] as did Pon1 depletion [12,14]. In the present work, Pon1 depletion by RNA interference or treatments with Hcy-thiolactone or *N*-Hcy-protein similarly increased the accumulation of Aβ in mouse neuroblastoma cells (Figure 5). Taken together, these findings suggest that increased accumulation of Aβ in Pon1-depleted brains is mediated by the effects of Hcy metabolites on mTOR signaling and autophagy. 

5xFAD mice develop sensorimotor deficits beginning at about 9 months of age (https://www.alzforum.org/research-models/5xfad-b6sjl) (accessed 27 December 2022). For example, 5xFAD mice perform worse than the wild-type animals in the hindlimb and balance beam tests [44,45]. We found that depletion of Pon1 did not aggravate these deficits: there was no difference in sensorimotor performance between *Pon1*^−/−^5xFAD mice vs. *Pon1*^+/+^5xFAD animals in the hindlimb, ledge, and cylinder tests (Appendix A). These findings suggest that upregulated Aβ accumulation may not be causing sensorimotor impairment. However, other aspects of sensorimotor abilities may be affected by Pon1, which remains to be assessed in future studies, as are the effects of Pon1 on various domains of cognition [24].

In conclusion, our findings define a mechanism by which Pon1 prevents Aβ generation in a mouse model of AD and neural cells.

## Figures and Tables

**Figure 1 cells-12-00746-f001:**
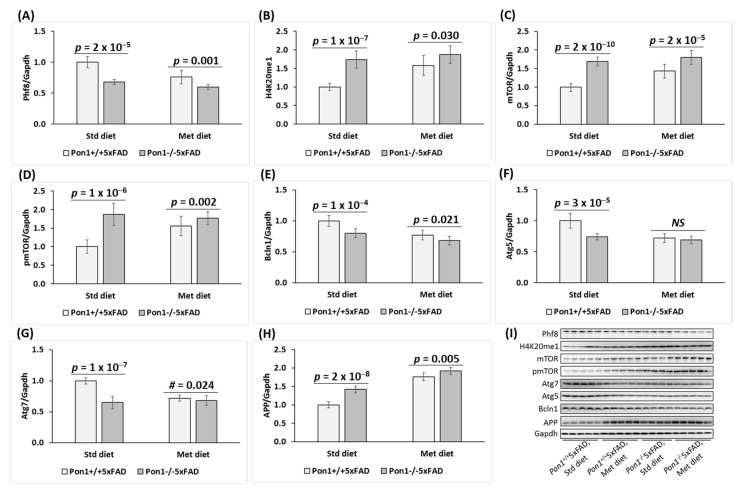
Pon1 depletion affects the expression of histone demethylase Phf8, histone H4K20me1 epigenetic mark, mTOR, pmTOR, autophagy-related proteins, and App in the *Pon1*^−/−^5xFAD mouse brain. (**A**–**H**) 5-month-old mice: One-month-old *Pon1*^−/−^5xFAD mice and *Pon1*^+/+^5xFAD sibling controls fed with HHcy high Met diet (1% Met in drinking water) or control diet for 4 months were used in experiments. Each genotype/diet group included 8–10 mice of both sexes. Bar graphs illustrating Western blot quantification of the following brain proteins are shown: Phf8 (**A**), H4K20me1 (**B**), mTOR (**C**), pmTOR (**D**), Bcln1 (**E**), Atg5 (**F**), Atg7 (**G**), and App (**H**). Representative pictures of Western blots are shown in panel (**I**). Gapdh protein was used for normalization. Data are averages of three independent experiments.

**Figure 2 cells-12-00746-f002:**
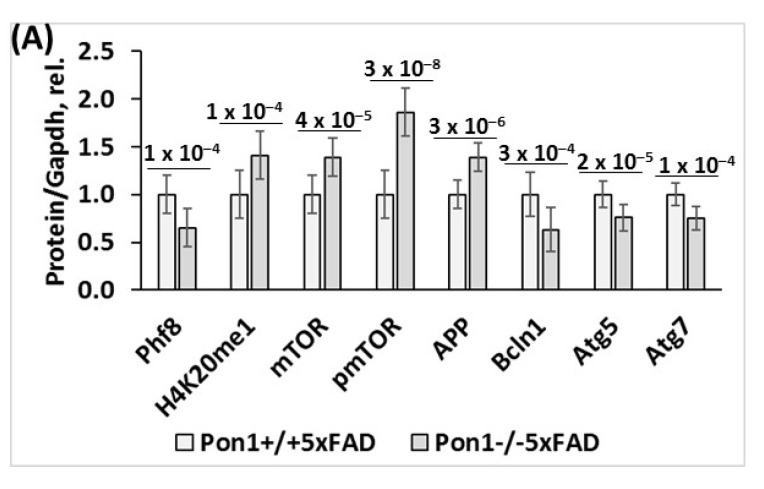
Pon1 depletion affects the expression of histone demethylase Phf8, histone H4K20me1 epigenetic mark, mTOR, autophagy, and App in the brains of 12-month-old *Pon1*^−/−^5xFAD mice. After weaning at 1 month, *Pon1*^−/−^5xFAD mice and *Pon1*^+/+^5xFAD sibling controls were fed with a standard diet for 11 month. Each genotype group included 10–12 mice of both sexes. (**A**) Bar graphs illustrate Western blot quantification of the indicated brain proteins. (**B**) Pictures of Western blots. (**C**) Bar graphs showing RT-qPCR quantification of mRNA for Pon1, Phf8, mTOR, autophagy-related proteins, and App. Gapdh mRNA levels was used as a reference for quantification. As expected, Pon1 was absent in the *Pon1*^−/−^5xFAD mice.

**Figure 3 cells-12-00746-f003:**
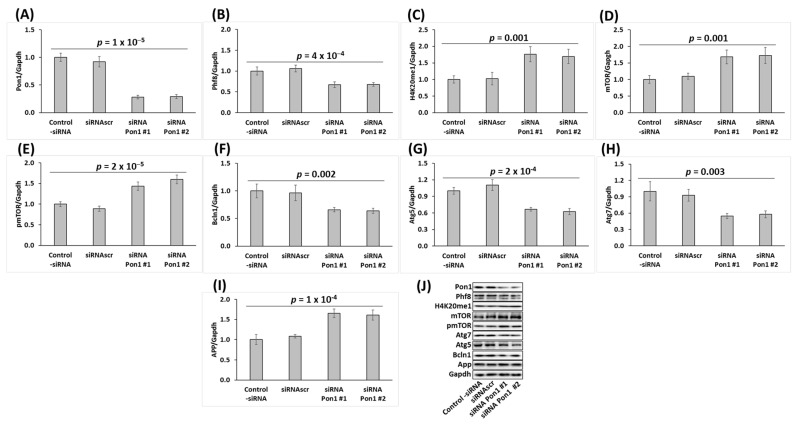
*Pon1* gene silencing in mouse neuroblastoma N2a-APPswe cells recapitulates changes in histone demethylase Phf8, H4K20me1, mTOR signaling, APP, and autophagy-related protein levels observed in *Pon1*^−/−^ mouse brain. Bar graphs illustrating the quantification of Pon1 (**A**), Phf8 (**B**), H4K20me1 (**C**), mTOR (**D**), pmTOR (**E**), Bcln1 (**F**), Atg5 (**G**), Atg7 (**H**), and App (**I**) in N2a-APPswe cells transfected with two different siRNAs targeting the *Pon1* gene (siRNA *Pon1* #1 and #2) are shown. Representative pictures of Western blots are shown in panel (**J**). Transfections without siRNA (Control -siRNA) or with scrambled siRNA (siRNAscr) were used as controls. Gapdh was used as a reference protein. Data are averages of three independent experiments.

**Figure 4 cells-12-00746-f004:**
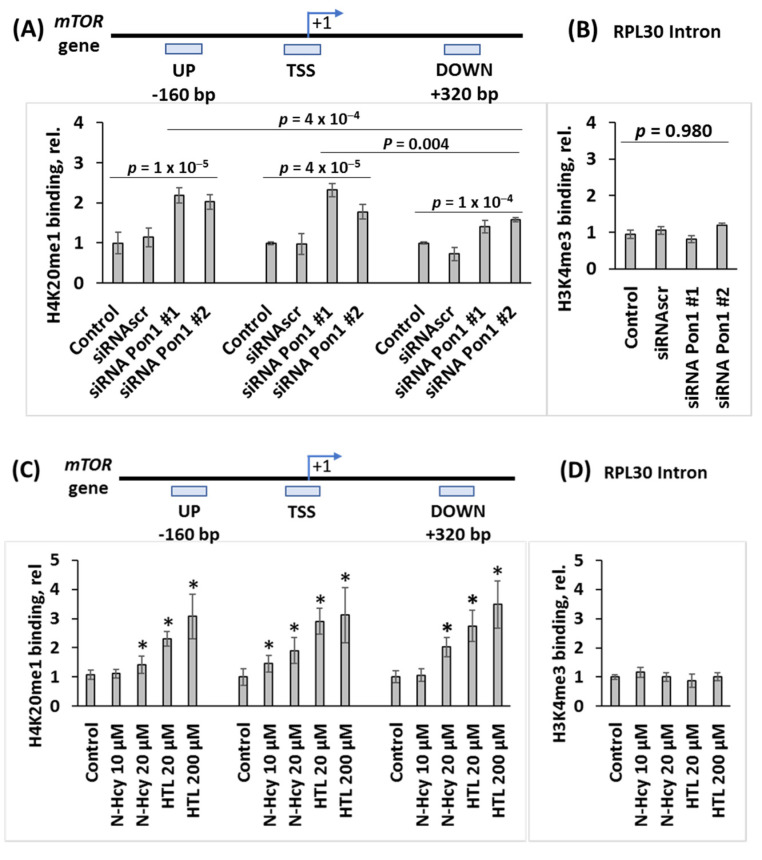
Pon1 depletion or treatments with Hcy-thiolactone or *N*-Hcy-protein increase H4K20me1 binding at the *mTOR* promoter in mouse neuroblastoma N2a-APPswe cells. CHIP assays with anti-H4K20me1 antibody show the specific binding of H4K20me1 at the transcription start site (TSS) of the *mTOR* gene as well as downstream and upstream sites. Bar graphs show the relative H4K20me1 binding at indicated regions of the *mTOR* gene. (**A**) N2a-APPswe cells were transfected with two different siRNAs targeting the *Pon1* gene (siRNA *Pon1* #1 and #2) (48 h, 37 °C). Transfections without siRNA (Control -siRNA) or with scrambled siRNA (siRNAscr) were used as controls. (**B**) Control CHIP experiment with anti-H3K4me3 antibody shows that *Pon1* gene-silencing did not affect the binding of H3K4me3 at the Rpl30 intron. (**C**) N2a-APPswe cells were treated with indicated concentrations of *N*-Hcy-protein or Hcy-thiolactone (HTL) (24 h, 37 °C). Untreated cells were used as controls. (**D**) Control CHIP experiment with anti-H3K4me3 antibody shows that Hcy-thiolactone or *N*-Hcy-protein did not affect binding of H3K4me3 at the Rpl30 intron. RT-qPCR was conducted on the input and precipitated DNA fragments. Data are averages of three independent experiments. * Significant differences vs. controls, *p* < 0.05.

**Figure 5 cells-12-00746-f005:**
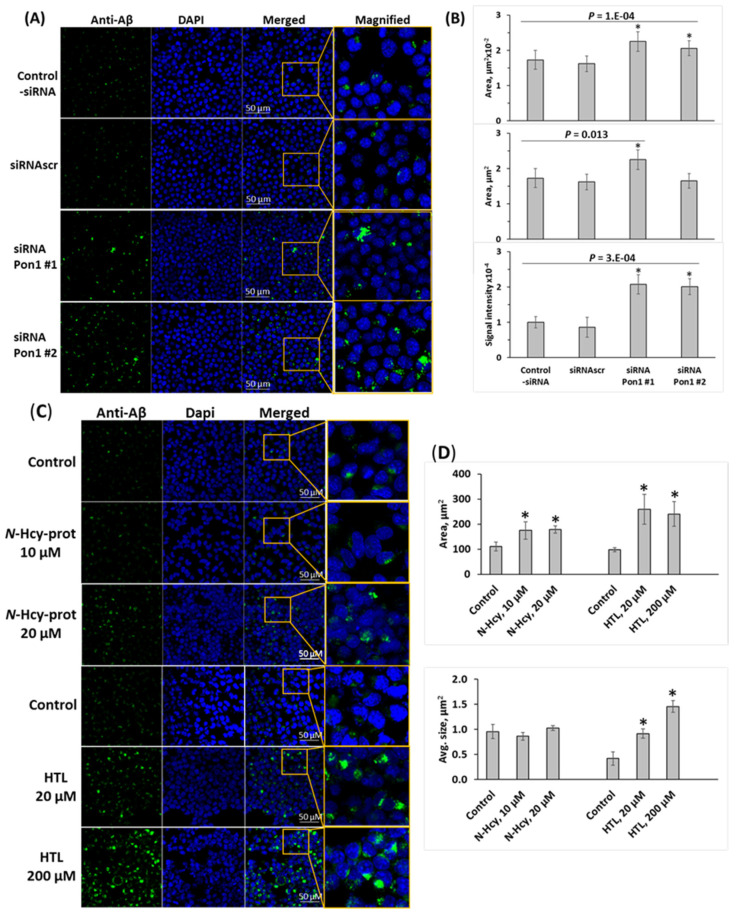
Pon1 depletion promotes Aβ accumulation in N2a-APPswe cells. (**A**–**D**) Analysis of Aβ in mouse neuroblastoma N2a-APPswe cells by confocal immunofluorescence microscopy using anti-Aβ antibody. (**A**,**B**) The cells were transfected with siRNAs targeting the *Pon1* gene (siRNA Pon1 #1 and #2). Transfections without siRNA (Control -siRNA) or with scrambled siRNA (siRNAscr) were used as controls. Confocal microscopy images (**A**) and quantification of Aβ signals (**B**) from *Pon1*-silenced and control cells are shown. (**C**,**D**) N2a-APPswe cells were treated with indicated concentrations of *N*-Hcy-protein or Hcy-thiolactone (HTL) for 24 h at 37 °C. Untreated cells were used as controls. Each data point is an average of three independent experiments with triplicate measurements in each. * Significant difference from control, *p* < 0.05.

**Figure 6 cells-12-00746-f006:**
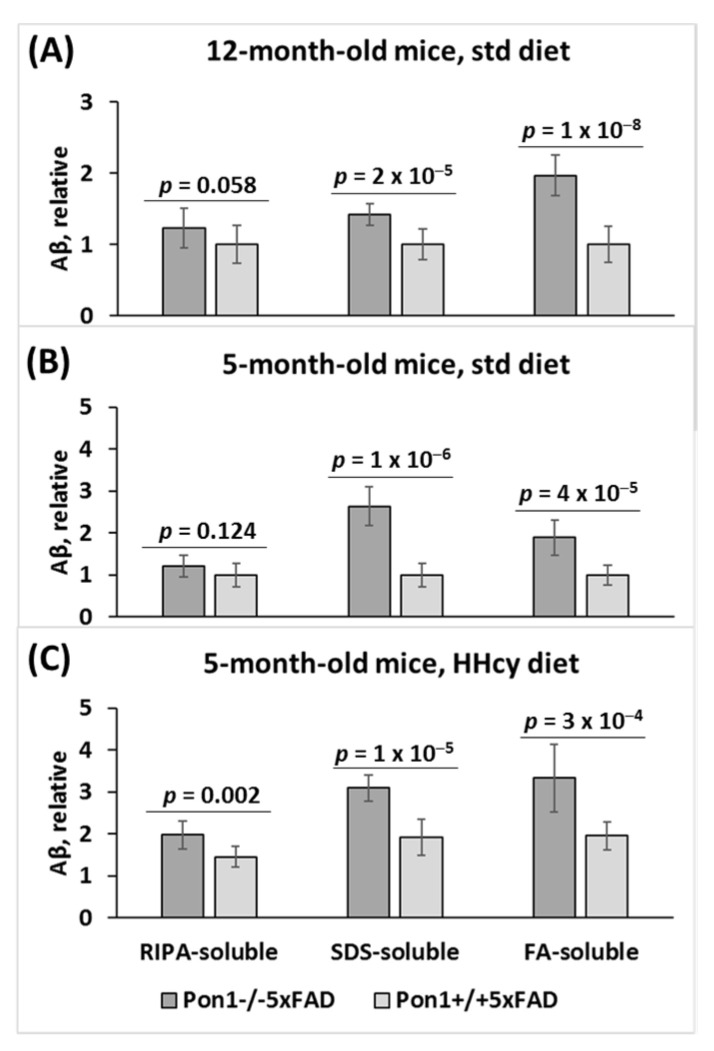
Pon1 depletion promotes Aβ accumulation in *Pon1*^−/−^5xFAD vs. *Pon1*^+/+^5xFAD mice. Aβ was quantified in RIPA-soluble, SDS-soluble, and FA-soluble fractions extracted from mouse brain. Twelve-month-old (**A**) and 5-month-old mice (**B**) fed with a standard diet as well as 5-month-old mice fed with a HHcy diet (**C**) were used in the experiments. Each measurement for an individual mouse was repeated three times. Aβ values shown are averages of measurements for 8–10 mice/group.

## Data Availability

The data that support the findings of this study are available in the methods and/or Appendix A of this article.

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
