# Peer review of "Depletion of Paraoxonase 1 (Pon1) Dysregulates mTOR, Autophagy, and Accelerates Amyloid Beta Accumulation in Mice"

_cells, 2023, doi:10.3390/cells12050746_

Round 1
Reviewer 1 Report
This paper proposed by Łukasz Witucki et al, entitled "Depletion of paraoxonase 1 (Pon1) dysregulates mTOR, autophagy, and accelerates amyloid beta accumulation in mice" focused on the Pon1 depletion affects the histone demethylase Phf8, mTOR signaling/autophagy, and amyloid beta (Aβ) accumulation in N2a-APPswe cells and in brains of Pon1-/-5xFAD vs. Pon1+/+5xFAD mice. Although most data presented are not very novel, this paper is interesting. However, several important points have to be addressed and mistakes have to be corrected before a publication could be considered.
1) Concerning ChIP experiments: Time of fixation should be added. What about quenching? how cells/nuclei were sonicated? More importantly, the negative control was done without control antibody that is not acceptable for a ChIP. Indeed, most of antibodies have a severe background noise and signal should be calculated in regards of this baseline not with no antibody condition. This experiment has to be repeated with IgG for negative IP.
2) The authors must include the concentration of L-Hcy-thiolactone or N-Hcy-protein used in the culture?
3) In figure 1, the authors showed upregulation/ downregulation of Phf8, histone H4K20me, mTOR signaling, autophagy, and App in the mouse brain. Please provide western blot images for all the genes.
4) In the results section the authors discuss about Figure S1, I did not find Figure S1 in the paper. The authors cited this figure number – 224,235,255,266,404,406,413,419,421.
5) The introduction needs to be elaborated citing the link between Pon1 and Autophagy.
6) The whole manuscript needs language editing in order to make it easier to read and follow. Please, stay consistent with the style.
Author Response
Responses to Reviewers
We thank the Reviewers for their comments which we used to improve our manuscript. Detained responses to the comments and the corresponding changes in the manuscript are described below.
Reviewer 1
Comment 1) Concerning ChIP experiments: Time of fixation should be added. What about quenching? how cells/nuclei were sonicated? More importantly, the negative control was done without control antibody that is not acceptable for a ChIP. Indeed, most of antibodies have a severe background noise and signal should be calculated in regards of this baseline not with no antibody condition. This experiment has to be repeated with IgG for negative IP.
Response: Please note that we used CUT&RUN Assay Kit #86652 (Cell Signaling Technology, 133 Danvers, MA, USA), which uses a proprietary technology that does require cell fixation step. We used a negative control - Rabbit (DA1E) mAb IgG XP® Isotype - included in the CUT&RUN kit, which did not afford any signals in RT-qPCR assays targeting mTOR. We have now clearly described this assay in section 2.7.
Comment 2) The authors must include the concentration of L-Hcy-thiolactone or N-Hcy-protein used in the culture?
Response: These concentrations are now included in section 2.4.
Comment 3) In figure 1, the authors showed upregulation/ downregulation of Phf8, histone H4K20me, mTOR signaling, autophagy, and App in the mouse brain. Please provide western blot images for all the genes.
Response: We have now included representative Western blots in panel I of Figure 1 and panel B of Figure 2.
Comment 4) In the results section the authors discuss about Figure S1, I did not find Figure S1 in the paper. The authors cited this figure number – 224,235,255,266,404,406,413,419,421.
Response: This was a mistake on our part, thank you for catching it. We have now corrected figure numbers.
Comment 5) The introduction needs to be elaborated citing the link between Pon1 and Autophagy.
Response: The Introduction section has now been rewritten and expanded to highlight the link between Pon1 and Alzheimer’s disease. However, we are not able to find any information regarding the link between Pon1 and autophagy. To our best knowledge, our present manuscript is the first to identify such a link.
Comment 6) The whole manuscript needs language editing in order to make it easier to read and follow. Please, stay consistent with the style.
Response: We have carefully checked the spelling, grammar, and writing style using the editor function in Microsoft Word. Our Editor Score is 100%.
We have also added new Figure 2 and information on mRNA quantification in Pon1-/-5xFAD mice (new sections 2.6 and 3.4), on Aβ quantification in Pon1-/-5xFAD mice (new section 3.8).
We have provided additional data in the Supplementary Material file containing
Supplementary Figure S1, Supplementary Figure S2, Supplementary Table S1.
Reviewer 2 Report
I think the conclusion is too brief and since the authors suggest they are preparing more manuscripts in this field and several aspects of AD are not studied, the conclusion that this paper completely addresses the mechanism of effect of Pon-1 deletion on amyloid beta and AD may not be correct.
Author Response
Responses to Reviewers
We thank the Reviewers for their comments which we used to improve our manuscript. Detained responses to the comments and the corresponding changes in the manuscript are described below.
Reviewer 2
Comment: I think the conclusion is too brief and since the authors suggest they are preparing more manuscripts in this field and several aspects of AD are not studied, the conclusion that this paper completely addresses the mechanism of effect of Pon-1 deletion on amyloid beta and AD may not be correct.
Response: To strengthen our conclusion regarding the involvement of Pon1 in protecting against Aβ accumulation, we have included additional data in Figure 2, Figure 5, and Supplementary Material
Comment: English language and style are fine/minor spell check required
Response: We have carefully checked the spelling, grammar, and writing style using the editor function in Microsoft Word. Our Editor Score is 100%.
Round 2
Reviewer 1 Report
The authors have provided updated the figures and included all the necessary information as per the comments. It is very well written and i would recommend publishing it.